# The Discussion of Potential Care Needs for Physically and Mentally Disabled Citizens in Taipei City by Using Spatial Analysis



**Jui-Hung Kao [1], Wei-Chen Wu [2,*], Cheng-Hu Chow [3] and Horng-Twu Liaw [1]**

[1] Department of Information Management, Shih Hsin University, Taipei 116005, Taiwan; kjhtw@mail.shu.edu.tw (J.-H.K.); htliaw@mail.shu.edu.tw (H.-T.L.)
[2] Department of Finance, Feng Chia University, Taichung 407802, Taiwan
[3] Department of Public Policy and Management, Shih Hsin University, Taipei 116005, Taiwan; nathan@mail.shu.edu.tw
[*] Correspondence: wwu@mail.fcu.edu.tw; Tel.: +886-4-2451-7250 (ext. 4165)

**Abstract:** What this research may achieve points towards the need to progressively improve the reasonableness in establishing Social Welfare Agencies (SWAs). The service capacity of SWAs is far below the population of the level III extremely disabled. This is a serious problem. This evaluation can assist social welfare and public health departments to determine what locations to approve for establishing SWAs in the short term and plan for new SWAs more precisely, as well as rein in budgetary priorities. As an illustration, in considering the distance between SWAs and the extremely disabled, the service quality of SWAs and fairness in the planning have to be taken into account. Introducing a Service Quantity Needed-Index for SWAs (SNIS) into the current measure of approving and planning new SWAs shall assist the departments in distributing social welfare resources to areas most in need of help. In addition, using the modified data to recalculate SNIS can examine needs regularly. Employing basic statistical areas for short-term applications in Taipei City SWA projects, considering the distance between SWAs and the extremely disabled, the agencies' service quality and fairness in the planning of SWAs need to receive more attention. Previous research mostly employed straight-line distances rather than road distances. To a certain extent, this overlooked the actual capacity of roads as well as led to some degree of discrepancies in evaluations. This essay focuses on calculating SNIS, mainly towards guiding the establishment of facilities and concretely proposing how to optimize their locations. Future research can add in needs at that time in accordance with current evaluation results to propose plans to optimize the locations, or maybe integrate weights of disability to adjust multiple requirements of SWAs.

**Keywords:** social welfare agencies; disabled people; weighted population center points; relaxed variable kernel density estimator

## 1. Introduction

The degree of aging (inhabitants 65 years old or above) of Taipei City is even more apparent than in other developed nations. Overall, the total number of elderly care centers in Taipei City is far below the needs of the elderly population. The term "crippled" was often used for people whose bodies or limbs had defects and thereby had lost their function. However, since this word has connotations of belittling and condemnation, in modern times it is replaced by "physically disabled". According to stipulations in the Physically and Mentally Disabled Citizens Protection Act passed by Taiwan's Legislative Yuan in 1997 [1], people eligible for an ID paper for the disabled include the visually disabled, disabled in hearing functions, balance functions, voice or language functions, limbs disabled, mentally disabled, and those whose important organs have lost function. Those with disfigured complexions, the vegetative, dementia patients, autism patients,

chronic psychiatric patients, the multiply disabled, patients of retractable (intractable) epilepsy, those who have been ascertained by the central health authorities to be disabled by rare illnesses and others are ascertained by the said health authorities to be disabled. Within the disabled population, the proportion occupied by those with limb disabilities is the highest, and the situation is also the most complex [2].

Regarding supply and demand in social welfare agencies, the degree of disability is classified into three levels. The first is lightly disabled, with Activity of Daily Living (ADL) Index evaluated scores of 61 to 80, or 81 and above with Instrumental Activity of Daily Living (IADL) evaluation for shopping and getting out, cooking, household work and laundry, more than two among the four requiring assistance. The second is medium to heavily disabled; the Barthel Index evaluated score is 31 to 60. The third is extremely disabled (ADL evaluation score 30 or below). Classified according to a state of health and length of time requiring care, the larger number means the longer the time needing care. In principle, the third type—extremely disabled—tends to choose social welfare facilities in familiar environments near to home. The reasons are that their living conditions are relatively difficult and the level of physical strength is lower. Service resources are distributed by the Taipei City Government. Methods of distribution consider the number of the disabled that Social Welfare Agencies (SWAs) can accommodate in consideration of the proportion of the disabled and the population of the extremely disabled in the area. The Service Quantity-to-Disability Population Ratio (SDPR) method is used on an SWA and the zone where the agency can provide service. The population of the extremely disabled is used for the said zone mapping basic statistical areas. These two methods define from above three levels of welfare resources, SWAs and zones where agencies can serve. However, they lack SDPR estimation of basic statistical areas.

This essay focuses on calculating SNIS, which is mainly towards guiding the establishment of facilities and concretely proposing how to optimize their locations. In addition to using the Two-Step Floating Catchment Area to find out the lack of resources, this research also supplemented with RVKDE and Mean Center to find the SWA with the most lack of resources region. To propose a plan for optimizing the location, or to integrate disability weights to adjust the various requirements of SWA.

## 2. Related Works

In order to tackle the needs of reasonably establishing SWAs by stages, four basic steps illustrate our method, variables (parameters) and analytical logic concerning the evaluation of the distribution of SWAs. Below we shall delineate the steps towards achieving the final goal of evaluating the distribution. First, Two-Step Floating Catchment Area (2SFCA) is introduced to consider the parameters of the distance between different SWAs [3]. We also added in the evaluation of SWA distribution at the level of the basic statistical area. Then through employing the potential need coefficient of the level III extremely disabled in the beginning parameters, we further improved the starting service catchment area of SWAs. We summed up the deficiencies of the existing method (SDPR), already mentioned, to resolve the needs to better reasonably establish SWAs.

Network analysis is introduced to consider the distance parameters between SWAs and the extremely disabled. In addition, we propose evaluation of the distribution of SWAs at the basic statistical area level. Then, through applying the potential need coefficients of the extremely disabled in the original parameters, we further improved the original SDPR method, which is less based on enhancement from measurement of the needed parameters integrating service catchment area and basic statistical area. Employing spatial location information regarding SWAs, integrating other data related to the agencies and putting together data concerned (e.g., total population, household numbers, elderly population, disabled population, medium- and low-income population) of the groups in need, currently and in the future 5–10 years, we evaluated the spatial accessibility of facilities for public service and social service, making use of the current 97 agencies, employing the spatial methods. The model framework as proposed by this essay is illustrated in Figure 1.

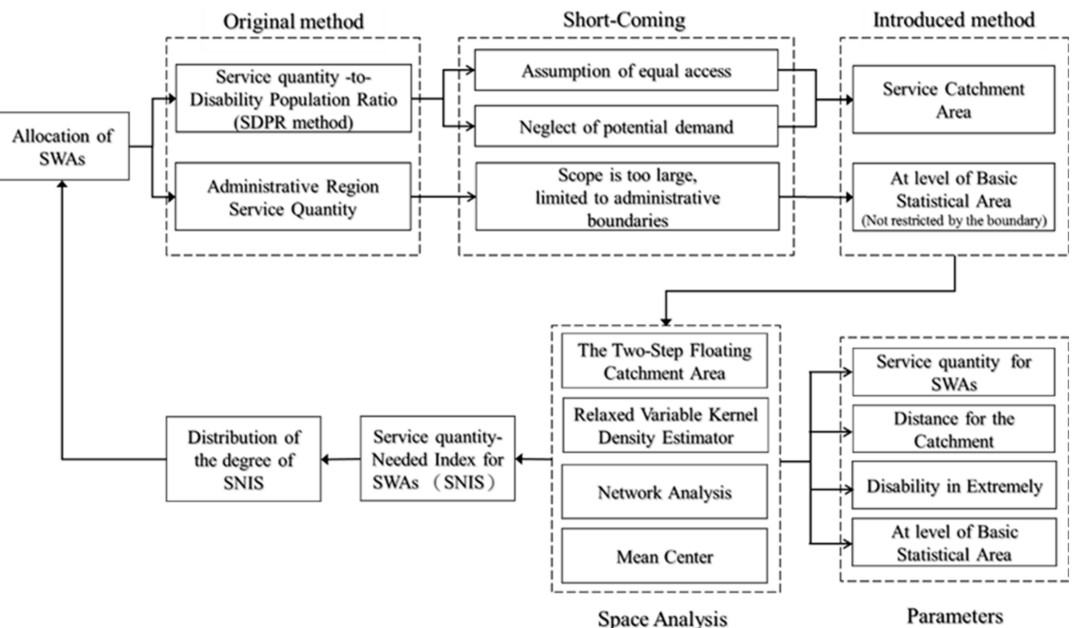

**Figure 1.** Framework chart.

## 3. System Modeling

After applying spatial analysis to system modeling, in spite of the fact that many scholars in the past used The Two-Step Floating Catchment Area (2SFCA) to find out the resource-deficient regions, they did not find the hotspots. In this study, Relaxed Variable Kernel Density Estimator (RVKDE) and Mean Center are used as the basis for determining the potential future needs of SWA, which can analyze the possibility of spatial accessibility evaluation for the population needing long-term care in each basic statistical area.

### 3.1. Two-Step Floating Catchment Area (2SFCA) Method

The evaluation of grassroots medical resources is mainly self-driving for 30 min or 15 miles as the reasonable usable zone [4]. However, the traffic environment of Taiwan has significant differences with that of the United States. This is due to large traffic volume on roads in Taiwan and the roads are generally narrower. The actual vehicular speed of traffic in Taiwan is often slower. The actual life activity zone is thereby also slimmer. Thus, this research refers to studies by scholars. Walking comfortably has the speed of about 1.22 m/s. It takes about 13 min to walk 1 km. For related factors influencing service satisfaction [5,6], there is the best satisfaction for approachability when the residence and service-providing unit has a distance "under 30 min". Hence, the optimized distance for this research is 2 km, about 26 min. The 2 km zone reachable by roads from the center of the sphere of the catchment area is the reasonable-use zone of social welfare resources for that sphere. Calculations with the two-step floating catchment method will be conducted.

The two-Step Floating Catchment Area (2SFCA) method proposed and broke through the limitations of activities framed by administrative borders as aforementioned [7]. The method does not just consider the possibility of populations seeking medical service across districts, but also establishes a reasonable medical-service seeking zone, so as to evaluate spatial accessibility of medical resources. This method is mainly divided into these two steps [4,8]:

1.  For each SWA, the possible service population covered within the reasonable service zone (e.g., 2 km) is searched, in order to come up with the provider-to-population ratio. What is shown in Figure 2 is the simple concept diagram of a two-step floating search method. It includes three SWAs (A, B, C) represented by village population centers and 15 population points of the catchment areas of the SWAs of the sphere. Using administrative region A as an example, population (1, 2, 3, 4, 6, 7, 9, 10)

within the 15 km distance zone is eight. Hence, the social welfare resource ratio in administrative region A is 1/8 (each SWA can serve eight disabled). The ratio is 1/4 for administrative region B (4, 5, 8, 11). That for administrative region C (11, 13, 15) is 1/3.

2. For each population point of potential needs, we search for possible social welfare providers within the reasonable service zone (e.g., 2 km) and add that to the ratio obtained from the perspective of the service provider in step 1. That will be the social service accessibility index we desire. Extending 15 km to search for possible social welfare resources from each sphere of the catchment area center point, results indicate that for 1, 2, 3, 6, 7, 9, 10 there is just A, the single SWA within 15 kms. Thus, their provider-to-population ratios are all 1/8. Similarly for 5, 8, 11, within the 15 km extended zone there is just B—the SWA. Their ratio is 1/4. But the sphere of catchment area for 4 is very special. It is because for the disabled living there, two SWAs-A and B—are included in their 15 km extension zone. Thus, the provider-to-population ratio for 4 is 1/8 + 1/4 = 3/8. Calculated according to the traditional administrative region method, the ratio of the disabled living in village 4 would be 1/6 from the perspective of the administrative region B (including 4, 5, 8, 11, 13, 15). If considered from the level of the sphere of catchment area (sphere of catchment area 4) the ratio is 0.

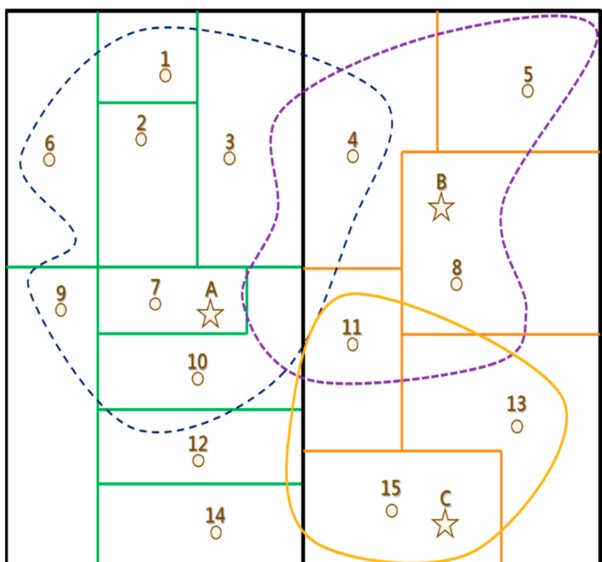

**Figure 2.** The Two-step Floating Catchment Area (2SFCA) Method.

*3.2. Relaxed Variable Kernel Density Estimator (RVKDE) Method*

Relaxed Variable Kernel Density Estimator (RVKDE) was jointly developed [9,10], and the basic concept of RVKDE is a random independent selection of sample (kernel point) sets in a given data distribution and using them to construct an estimation function close to the primary data. Suppose the primary data is distributed in a feature space of m dimensions according to kernel density function *f*, using RVKDE the algorithm will be the sample sets {s1, s2, . . . sn} as kernel points. According to data aggregation close to each kernel point, each of their kernel functions will be estimated. Lastly, RVKDE conducts linear-weighted summation of all kernel functions to arrive at a probability density function $\hat{f}$ to estimate *f*. For spatial data point *v*, the output of function *f* can be estimated as:

$$\hat{f}_j(v) = \frac{1}{|s_j|} \sum_{S_i \in S_j} \left( \frac{1}{\sqrt{2\pi \cdot \sigma_i}} \right)^m exp \left( -\frac{||v - s_i||^2}{2\sigma_i^2} \right) \qquad (1)$$

while $\sigma_i = \delta_i \times \beta$, $\delta_i = \frac{\overline{R}(s_i)\sqrt{\pi}}{\sqrt[m]{(k+1)\Gamma\left(\frac{m}{2}+1\right)}}$, $\overline{R}(s_i) = (m+1/m)\left((1/k)\sum_{h=1}^{k_1}||\hat{s}_h - s_i||\right)$, $R(s_i)$ is the maximum distance of kernel point $s_i$ and neighboring $k$ data points; $\Gamma(\cdot)$ represents Gamma function; $\beta$ and $k$ are parameters to be cross-examined or set by user.

### 3.3. Network Analysis Method

The network maze of network analysis is a circular shaped functionality, composed of lines and nodes linked together. It has been applied in the real world such as streets, railway, public channels, rivers and traffic systems. There are two types of networks in ArcGIS. One is channel and the other is a street. Network relationships have to be established for both to allow analysis. The network relationship of the channels is called geometric network of streets and network data set. This research focuses on street network analysis in order to organize distribution rules of the social welfare service sector and human resources. Thus, there are six tools related to ArcGIS network analysis [11,12]:

1. New Routes: It is used for noticing all the obstacles between position A and position B, then further calculates the best route.
2. New Service Zones: Used for searching all services around any position in the network. The zone includes all streets involved with obstacles set. This module can calculate all streets that aid services can reach in 10 min or within 10 kms.
3. Nearest New Facilities: The module is used for searching nearest hospitals when an incident occurs, to find the police or the closest shop nearest to the crime scene. The no. of search results is adjustable. It can also indicate the best route and calculate the transportation cost together.
4. New Origin Destination (OD) cost array: It is an array formed from multiple initial positions. Through cross-calculation this module will build a list according to cost and store this list in the attribute diary. Then it can indicate the cost to the destination from any point.
5. New Vehicle Route Problem (VRP): Under conditions such as no. of trucks, their load, time limit and priority levels, the module is often used by shipment centers to calculate the best delivery route. VRP aims to solve any complicated distribution problems as well as analyze lower costs.
6. New Location Distribution: When a shop owner attempts to reduce the no. of outlets, the owner must be concerned about not impacting customer demands when closing specific outlets in a specific area. When an aid team has been set up, this method will calculate and find out the best position for setting up a second team. Thus, this method can minimize the zone between needs and operation location as well as maximize the number of requests within the area of operation.

Using network analysis to compute instructions in the service area of research on the zone of service of resource locations for the disabled, compared to the zonal distance set according to the objects under buffer analysis, network analysis can precisely draw a road network in the zone defined by the researcher.

### 3.4. Mean Center Method

Mean Center approaches all factors' average x coordinate, y coordinate and z coordinate in the researched zone. Mean center is very useful for analyzing and tracking changes in distribution and comparing the distribution of factors of different types. For describing quantities of statistics on spatial distribution, using Mean Center point or Median Center point to describe centralizing trends, the distribution center is an important parameter for discrete points distributed along a surface. This can roughly indicate how the entire distribution is located.

In general, the geometric center of the set of points will be the position sought. Thus, this research follows such a method with the Mean Center functionality to discover the geometric center's position for this set of points as in Figure 3. This location will be the

most possible hot zone. Each point's *x* and *y* coordinates will be averaged as the calculation pivot. It can be represented as:

$$\overline{X} = \frac{\sum_{i=1}^{n} X_i}{n}, \quad \overline{Y} = \frac{\sum_{i=1}^{n} y_i}{n} \tag{2}$$

$x_i$ and $y_i$ are the *x* and *y* coordinates of a certain point *i*. *n* is the weighted average center of the totality of data points. It can be expanded as:

$$\overline{X}w = \frac{\sum_{i=1}^{n} w_i x_i}{\sum_{i=1}^{n} w_i}, \overline{Y}w = \frac{\sum_{i=1}^{n} w_i y_i}{\sum_{i=1}^{n} w_i} \tag{3}$$

$w_i$ is the weight of factor *i* [13].

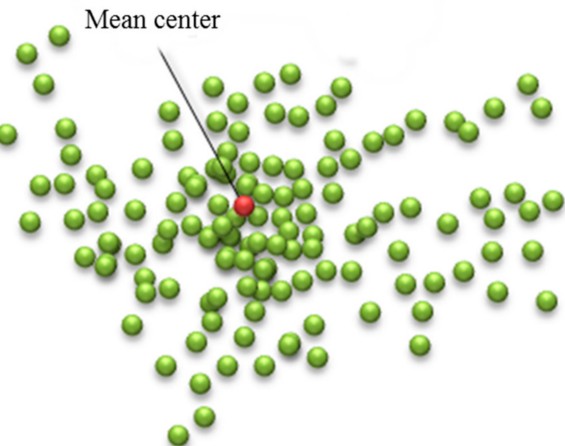

**Figure 3.** Diagram indicating the geometric center location of the Mean Center group.

Regarding crime, hotspot analysis methods can be used to study the distribution of common burglaries and their relations with activities around, to contrast and evaluate the points of incidents in daytime and at night, to study if the average center of common burglaries changes, helping law enforcement to better distribute resources [14]. In emergency medicine, comparison of the average center for emergency 110 calls and the locations of responding emergency stations can be used for evaluating states of service, to gauge new methods for the usage of potential caring facilities [15,16].

In Banff National Park, it has been used for evaluating quality of habitats for bison and estimating if that is enough for the supported groups, furnishing a useful model for evaluating introducing other endangered species into potential habitats of their original territories.

## 4. Analysis

This research makes use of descriptions from Government Website Open Information Announcement (DATA.GOV.TW) about the dataset "Statistical Areas Classification System" constructed by the Department of Statistics, Ministry of the Interior. The significance for establishing the basic statistical area lies in these. Although particular data can be used as the fundamental unit in any statistics, specific issues concerning privacy, sensitivity, and so on would mean most data cannot be publicly released. Thus, they could only be used by the responsible authorities or users with specific purposes. In the past, when it came to data gathering, organization and analysis, statistical units in Taiwan were largely based on administrative regions such as counties, municipalities, villages, and urban districts. Statistical analyses cannot be further conducted on refined small zones or specific territories. Administrative regions would have their borders adjusted, be combined or redrawn out of administrative needs. It leads to time sequence inconsistency in data units, reducing analytical gains of the data under a time sequence. When long-term systematic fixed small or specific zones are established as the smallest spatial units, particularly for statistical

usages, protecting privacy of the cases and upon further integration, endowing spatial characteristics of various socio-economic data at all statistical levels, it will be advantageous to the strengthening and deepening of spatial analyses on socio-economic data, raising their intrinsic values and application potentials.

### 4.1. Estimation at Basic Statistical Areas and the Distribution of the Degree of SNIS

The zone reachable by road as extended 2 km from a basic statistical area is considered to be the reasonable SWA approachable zone for the level III extremely disabled. Our aim is to make use of road network analysis to calculate the service population possible for the SWAs, to estimate their average service resources, and then deduce the aggregate of surrounding SWA service resources each basic statistical area can obtain. This research employs statistical data of related resources of the SWAs provided by the Department of Social Welfare, Taipei City, to estimate the amount of service clients in effect possibly rationed to each basic statistical area within 2 km of road access. There are 16,136 extremely disabled in Taipei City. From road network analysis 15,052 can be served within 2 km of road access by the 97 SWAs. There are 1084 persons without SWAs within 2 km from where they are. They are in Nangang, Xinyi, Beitou, Wenshan, Neihu, Zhongshan, and Shilin are shown in Figure 4.

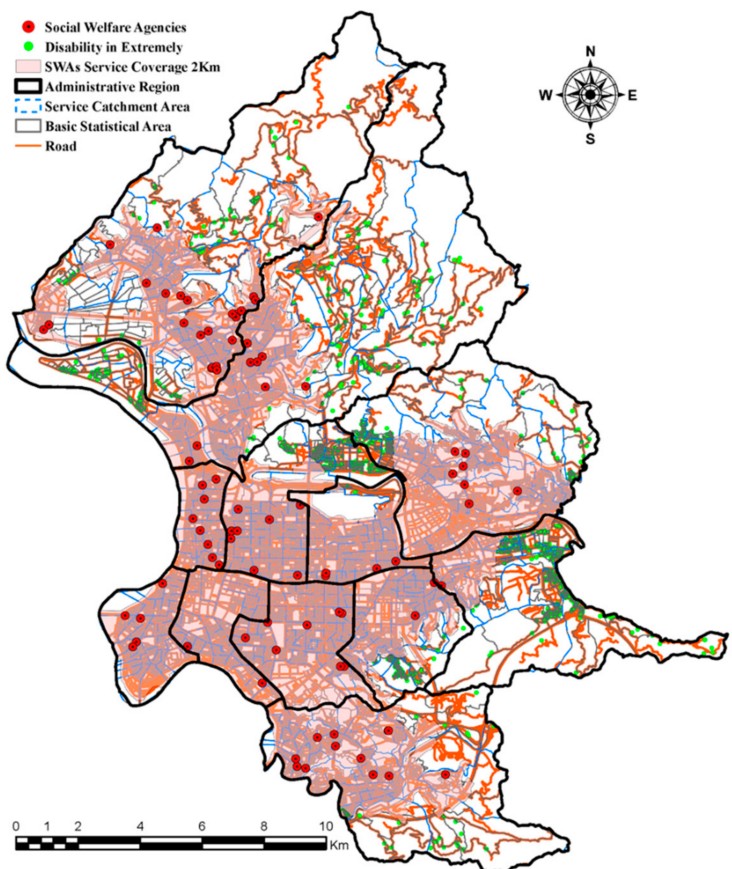

**Figure 4.** 2 km Road Network Map of Social Welfare Agency (SWA) Service Provision in Taipei City.

### 4.2. Analysis of Non-Approachability to Social Welfare Resources for the Extremely Disabled

Since most previous approachability research is applied to medical resources, this research commences from the definition of approachability in medically related research and attempts to derive our own possible definitions. The definition of approachability in the previous literature is actually very heterogeneous and variegated. From literature in the early 1980s pinpointing approachability of medical resources, the so-called concept of approachability was defined in a broader manner: "Approachability in a certain ex-

tent concerns the capability or the will of the users to enter the medical service system". Approachability is further defined as "degree of fit between the user and the medical system" [17–19].

Compared to accessibility, the concept of approachability is more related to space. Accessibility is more concerned with the proportion of possible persons in needs and resources that can be supplied in a fixed spatial zone. Thus, it means that in a specific zone, under the condition of more persons with needs and fewer resource providers, accessibility will be lower. On the other hand, approachability refers more to the spatial obstacles between users and service providers, for example, distance and time. When the distance is farther or traffic time longer, approachability between users and service resources will be lower, due to spatial obstacles becoming bigger. For the calculation in the second stage of the two-step floating search method of this research, namely through the spatial positions of the 16,136 level III extremely disabled populations, by road network analysis the 2 km road access zone is the reasonable approachable zone of SWAs, aiming to employ the possible service population of the SWAs calculated in stage one to estimate the service resources they can supply, and then deduce the SWA resource aggregate around each basic statistical area. This research makes use of statistical data of SWA-related resources provided by the Department of Social Welfare, Taipei City, to estimate within 2 km of road access for each basic statistical area of Taipei City the SWA resource that may be assigned to each of the extremely disabled, shown in Figure 5. The resources are mostly aggregated in Shilin, then Wanhua, Datong, Beitou, Wenshan, and Zhongshan. Districts lacking welfare agency resources are Songshan, Xinyi, and Daan.

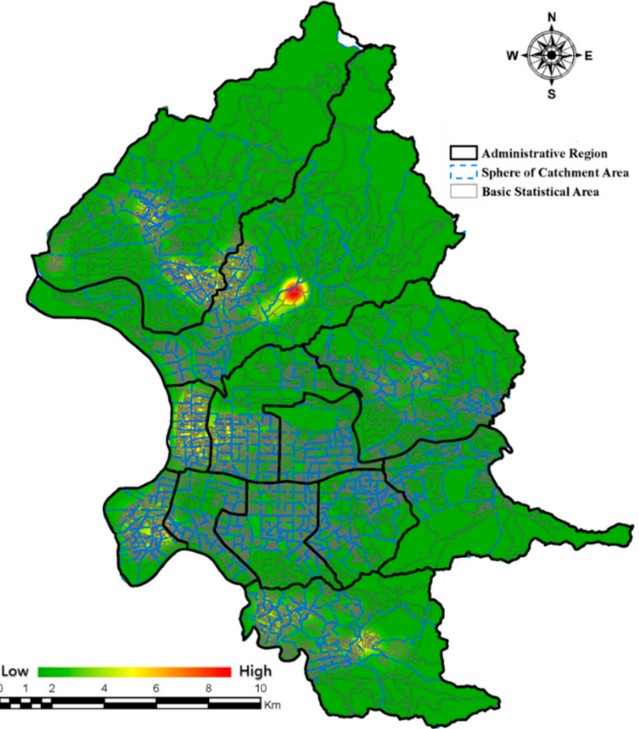

**Figure 5.** 2SFCA Map of SWA Service Provision in Taipei City.

### 4.3. Redressing Non-Approachability to Social Welfare Resources for the Extremely Disabled

After we have gathered and organized related data for analysis, as an example, we shall make use of the district with the greatest lack, which is Nangang. Spatial analysis will commence. Here it chiefly means evaluation of spatial approachability of SWAs. The method of analysis here is mainly according to the aforementioned research method of road network analysis and Relaxed Variable Kernel Estimator. In spatially redressing non-

approachability of the extremely disabled to social welfare resources, steps of executing the method are as below:

1.  Searching for 2 km zones that SWAs cannot serve—employing RVKDE combined with the 2 km road network map of SWA services in Taipei City as the basis, 2 km zones that SWAs cannot serve are found. We discovered Nangang has the greatest deficiencies, followed by where Zhongzheng and Neihu meet, shown in Figure 6a.
2.  Spatial analysis on 2 km zones that SWAs cannot serve and Mean Center—this research selected Nangang district with the greatest lack, as manifest by 2 km SWA nonservice zones, to conduct resource input spatial analysis. We input information such as population of the extremely disabled in the nonservice zones, the areas of the zones' hotspots and so on into active map layers for forthcoming statistical data conversion purposes. We made use of RVKDE and Create Contour on basic statistical areas with nonservice zones, pinpointing them to conduct mean-center analysis of the extremely disabled, to discover points for resource input; their basic statistical areas are shown in Figure 6b,c.

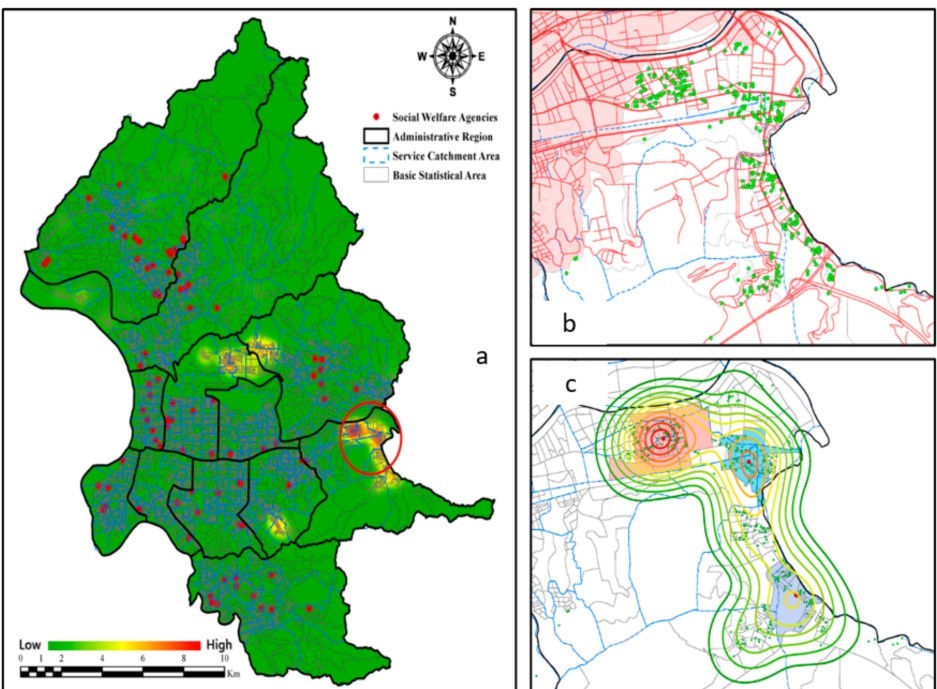

**Figure 6.** 2 km Road Maps of SWA Service Provision in Taipei City. (**a**) Hot zones of resource shortage map; (**b**) Resource shortage of spatial demand distribution map; (**c**) SWA space map of the best resource input.

## 5. Conclusions

This essay utilized data of basic statistical areas of Taipei City as a proposed idea to calculate SNIS. It employed network analysis to enrich the study with considerations between the distance of SWAs and the extremely disabled. It also conducted optimization measurements of the extremely disabled demographic group in need of SWAs. Regarding basic statistical areas having zones without service, Mean Center analysis was conducted on the extremely disabled population to discover points for resource inputs and their basic statistical areas. On the other hand, under limited means, social service resources in the reachable zones may not be able to satisfy those with the highest demands for them. Making use of 2SFCA to evaluate spatial approachability, we may also simultaneously discuss whether resources in the reachable zones are enough. Considering areas with relatively low SDPR, one may consider raising the service capacity of SWAs, or by referring

to non-approachability analysis of the extremely disabled to social welfare resources, increase the number of SWAs.

This essay designed a methodology to evaluate the input of spatial resources. The evaluation is by the spatial approachability of public service facilities and social service provisions. Through the tools of Two-Step Floating Catchment Area, RVKDE and Mean Center within this method as our basis, SWAs of future potential needs were discovered. The results were used to analyze the evaluation of the spatial approachability of the SNIS population of each basic statistical area, to serve government departments without enough budgetary outlay to supply the amount of social welfare service in need. Under a limited budget, locations to deploy social welfare services must be efficaciously selected. Using RVKDE, we can discover where are the priorities for deploying social welfare services. Furthermore, the same method can be used in each district, and this analysis on two levels provided government decision makers with more useful information.

**Author Contributions:** Conceptualization, J.-H.K.; data curation, C.-H.C.; formal analysis, W.-C.W. and H.-T.L.; methodology, W.-C.W.; resources, H.-T.L.; supervision, J.-H.K.; Validation, C.-H.C. All authors have read and agreed to the published version of the manuscript.

**Funding:** This research received no external funding.

**Institutional Review Board Statement:** Not applicable.

**Informed Consent Statement:** Not applicable.

**Data Availability Statement:** Not applicable.

**Conflicts of Interest:** The authors declare no conflict of interest.

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
