# Peer review of "The Discussion of Potential Care Needs for Physically and Mentally Disabled Citizens in Taipei City by Using Spatial Analysis"

_sustainability, doi:10.3390/su13052665_

Round 1

Reviewer 1 Report

Authors wrote an interesting paper with interesting anecdotes. Appreciate that the suggestions provide a new way to think of the density estimator that may help to improve the care and needs toward disabled people of varying degrees by the SWAs.

Only suggestions are:

-Perhaps to proofread the paper again. Make passive sentences active instead. Likely the original draft was written in another language (Chinese), and some translation was made from Chinese to English. So some areas are worded a bit awkward, like under 3.1 - 2SFCA Method:

"Regarding the evaluation of grassroots medical resources are mainly by self-driving for 30 minutes or 15 miles as the reasonable usable zone.  However, the traffic environment of Taiwan has significant differences with the USA's." 

-For Figures 3,4,5 and 6, I feel that some illustrations or examples to show how the figures could be used will be better. Right now, some figures seem to exist in isolation, or just put in-text, without the readers having knowledge of how they could be applied. So, illustrating with some numbers/pointers as examples on the figures themselves, and following up with some in-text description, to pinpoint the spots in the figures will draw the readers closer and help them understand what the figures are used for.  This exercise will also allow one to see how the formulas are being applied.

Overall, great contribution to adult care.

Author Response

Thank you very much for the recognition of the method mentioned in this review. We have revised the passive sentences to active. As for Taiwan’s long-term care is mainly for the elderly who can walk to the long-term care institution, it is very different from European and American countries. Rather than using automatic driving for 30 minutes or 15 miles as the reasonable distance in European and American countries,  figures 3, 4, 5, and 6 illustrate the visualized results of the space after the method of this research.

Reviewer 2 Report

The title is not appropriate and needs to be corrected. The methods dont have to be in the title. The focus of research is on the geographical availability with different methods. There is no hypothesis and results in the abstract.

The introduction is to general. It is necessary to define the concept (term) to availability. Which authors have conducted similar research? (look at references: Cabrera-Barona et al. 2015,2018; Wan et al. 2013; Šiljeg et al. 2018 (Annales)...

It is necessary to explain why these methods were chosen? What are the advantages or disadvantages for chosen methods and for another methods (average nearest neighbor, hot-spot analysis, cost-weighted distance methods...)

Author Response

Thank you for your suggestion. We gratefully appreciate your valuable suggestion and have revised the title part. Research on resource accessibility has been conducted for many years, and many scholars have devoted to related research. The difference between this research and other researches is not only through spatial accessibility to find out which areas lack resources, but also can further identify the hot spots of which resources are lacking, as well as to facilitate the effective use of limited resources.

Reviewer 3 Report

It is judged as a quantitative study based on empirical analysis through cases for the basis of the establishment of Social Welfare Agencies (SWAs). It is believed that the research structure is properly designed in the process of quantifying each qualitative evaluation item. The system modeling process is explained as shown in Figure 2 through the Two-Step Floating Catchment Area (2SFCA), and it is recommended that detailed explanations be added. In addition, it is judged that readability will increase if the process of what kind of process has been carried out for methodological selection prior to discussion of the Network Analysis Method. From the analysis results, the service area for SWAs was established through Taipei City. If so, major discussions and implications through mutual comparison between the existing SWAs and the results in the paper should be presented in the results in detail, and from these analysis results, the location of future SWAs. And the usefulness of how it can be used for the basis of establishment should be sufficiently summarized in the analysis implications.

Author Response

Thank you for your rigorous consideration. In this study, the social welfare organization has been established. The appropriateness of resource allocation and methods for spatial accessibility are to find out the region which lack of resources. In addition, we use the concept of SWA service area, while current government units use the administrative area as the resource input unit and the population determines the resource in general, rather than considering of existing demand. This study proposes to modify the social welfare SWA service area instead of the administrative area as the resource input unit and to achieve effective use of limited resources.

Round 2

Reviewer 2 Report

The title part is much better but it is necessary to explain why these methods were chosen (in one or two sentences). There is no results in the abstract and that is important. Write the research result in one sentence (in abstract).

Author Response

Thank you for your suggestion. The suggestion for the abstract has been revised in the content, and the reasons for choosing these methods have been revised and added to the System Modeling chapter.